# Electrophysiological Screening to Assess Foot Drop Syndrome in Severe Acquired Brain Injury in Rehabilitative Settings

**DOI:** 10.3390/biomedicines12040878

**Published:** 2024-04-16

**Authors:** Francesco Piccione, Antonio Cerasa, Paolo Tonin, Simone Carozzo, Rocco Salvatore Calabrò, Stefano Masiero, Lucia Francesca Lucca

**Affiliations:** 1Neurorehabilitation Unit, Section of Brain Injury Rehabilitation, Hospital-University of Padua, 35128 Padua, Italy; 2S. Anna Institute, 88900 Crotone, Italy; patonin18@gmail.com (P.T.); simone.carozzo@gmail.com (S.C.); l.lucca@istitutosantanna.it (L.F.L.); 3Institute for Biomedical Research and Innovation (IRIB), National Research Council of Italy, 00186 Messina, Italy; 4Pharmacotechnology Documentation and Transfer Unit, Preclinical and Translational Pharmacology, Department of Pharmacy, Health Science and Nutrition, University of Calabria, 87036 Arcavacata, Italy; 5IRCCS Centro Neurolesi “Bonino Pulejo”, 98124 Messina, Italy; roccos.calabro@irccsme.it; 6Neurorehabilitation Unit, Department of Neuroscience, University of Padua, 35128 Padua, Italy; stef.masiero@unipd.it

**Keywords:** critical illness polyneuropathy, critical illness myopathy, electrophysiological screening, acquired brain injury, rehabilitation outcomes

## Abstract

Background: Foot drop syndrome (FDS), characterized by severe weakness and atrophy of the dorsiflexion muscles of the feet, is commonly found in patients with severe acquired brain injury (ABI). If the syndrome is unilateral, the cause is often a peroneal neuropathy (PN), due to compression of the nervous trunk on the neck of the fibula at the knee level; less frequently, the cause is a previous or concomitant lumbar radiculopathy. Bilateral syndromes are caused by polyneuropathies and myopathies. Central causes, due to brain or spinal injury, mimic this syndrome but are usually accompanied by other symptoms, such as spasticity. Critical illness polyneuropathy (CIP) and myopathy (CIM), isolated or in combination (critical illness polyneuromyopathy, CIPNM), have been shown to constitute an important cause of FDS in patients with ABI. Assessing the causes of FDS in the intensive rehabilitation unit (IRU) has several limitations, which include the complexity of the electrophysiological tests, limited availability of neurophysiology consultants, and the severe disturbance in consciousness and lack of cooperation from patients. Objectives: We sought to propose a simplified electrophysiological screening that identifies FDS causes, particularly PN and CIPNM, to help clinicians to recognize the significant clinical predictors of poor outcomes in severe ABI at admission to IRU. Methods: This prospective, single-center study included 20 severe ABI patients with FDS (11 females/9 males, mean age 55.10 + 16.26; CRS-R= 11.90 + 6.32; LCF: 3.30 + 1.30; DRS: 21.45 + 3.33), with prolonged rehabilitation treatment (≥2 months). We applied direct tibialis anterior muscle stimulation (DMS) associated with peroneal nerve motor conduction evaluation, across the fibular head (NCS), to identify CIP and/or CIM and to exclude demyelinating or compressive unilateral PN. Results: At admission to IRU, simplified electrophysiological screening reported four unilateral PN, four CIP and six CIM with a CIPNM overall prevalence estimate of about 50%. After 2 months, the CIPNM group showed significantly poorer outcomes compared to other ABI patients without CIPNM, as demonstrated by the lower probability of achieving endotracheal-tube weaning (20% versus 90%) and lower CRS-R and DRS scores. Due to the subacute rehabilitation setting of our study, it was not possible to evaluate the motor results of recovery of the standing position, functional walking and balance, impaired by the presence of unilateral PN. Conclusions: The implementation of the proposed simplified electrophysiological screening may enable the early identification of unilateral PN or CIPNM in severe ABI patients, thereby contributing to better functional prognosis in rehabilitative settings.

## 1. Introduction

Foot drop syndrome (FDS) is a common condition in people with severe acquired brain injury (ABI). It is characterized by significant weakening and atrophy of the dorsiflexor muscles of the foot. In cases where the illness is unilateral, the underlying cause is typically peroneal neuropathy (PN), which arises from compression of the nerve trunk on the fibula’s neck at the knee level. Concomitant lumbar radiculopathy is less common. Bilateral and symmetric deficits of the muscles of the anterolateral compartment of the legs are caused by polyneuropathies and myopathies. This syndrome is mimicked by central palsies caused by lesions in the brain or spinal cord, which are typically accompanied by other symptoms including spasticity [1,2]. Individuals diagnosed with a traumatic brain injury (ABI) may experience peripheral FDS because of extended bed rest, the use of leg braces or casts, or improper leg placement that compresses the nerve at the fibula’s neck [3,4]. For this reason, passive mobilization and frequent changing of posture must be started in the first hours after admission to intensive care (ICU) or sub-intensive rehabilitation units (IRU). In addition to requiring mechanical breathing and lengthy treatments for multiple organ failure, patients with ABI in the intensive care unit typically require high dosages of anesthetics, corticosteroids, and antiseptics. For this reason, patients develop critical polyneuropathy (CIP) or myopathy (CIM), and often the two pathologies coexist (CIPNM) and are considered and studied as a combined clinical condition [5,6,7,8]. CIP and/or CIM weakens the leg muscles as well as the respiratory muscles, resulting in bilateral FDS, making weaning from artificial ventilation more difficult [4]. The diagnosis of PN, CIP and/or CIM is based mainly on electrophysiological tests, including nerve conduction studies (NCS) in conjunction with needle electromyography and repetitive nerve stimulation bilaterally performed on the upper and lower limbs. However, the whole examination in ICU and IRU is constrained by several factors, including the difficulty of the tests, the scarcity of neurophysiology consultants, and the patient’s lack of cooperation in cases of severe unconsciousness [9,10,11,12,13]. 

Previous studies suggested an alternative screening method, utilizing peroneal motor nerve conduction alone or in conjunction with sural sensory nerve conduction evaluation, as a simplified diagnostic NCS tool, primarily for PN or CIP, in place of performing the full electrophysiological tests, to reduce time and constraints for this neurophysiological examination [5,6,7,8,9]. According to Latronico et al. [11], CIPNM is predicted by an atypical decrease in the peroneal nerve’s compound muscle action potential (CMAP) amplitude. However, there were limitations to this screening technique for determining the muscle involvement associated with CIM. Nearly half of the patients in this study had an unclear diagnosis. For individuals who are not cooperative, direct muscular stimulation (DMS) combined with routine NCS testing may be an effective way to differentiate CIP from CIM [14]. The described method compared direct muscle stimulation CMAP (dmCMAP) to nerve-evoked CMAP (neCMAP). In CIM, the ratio between neCMAP and mCMAP was closer to 1, whereas in CIP the ratio was <0.5. Other authors have modified the technique, but the results remain the same [15,16,17].

The purpose of this study is to apply a simplified electrophysiological screening that may allow early and simpler identification of PN, CIP and/or CIM in patients with severe ABI in rehabilitative settings, thereby contributing to predicting specific clinical outcomes.

## 2. Materials and Methods

### 2.1. Study Design

Between 1 April 2023, and 30 November 2023, we conducted a prospective observational study on 28 individuals with severe ABI, unilaterally or bilaterally characterized by weakness and hypotrophy of the dorsiflexion muscles of the feet, who were admitted to the S. Anna Institute’s IRU in Crotone, Italy. To determine the prevalence of PN, CIP and/or CIM in a rehabilitative setting, patients were examined at admission using the proposed electrophysiological screening, in combination with a head and lumbar spine CT scan (see below). Inclusion criteria were: (a) age ≥ 18 years; (b) ABI including traumatic and vascular injury with a severe disorder of consciousness (DoC) of variable duration. Subjects with post-anoxic brain injury etiopathogenesis, pre-existing peripheral nervous system disorders and/or concomitant lumbar radiculopathies were excluded from the study. Following these criteria, only 20 subjects were included in the study.

During the hospital stay, a comprehensive multidisciplinary rehabilitation program was provided to all patients. Rehabilitation and pharmacologic interventions were homogeneously planned over the study period according to patients’ needs.

The study was carried out following the rule of the Declaration of Helsinki, approved by the local Ethic Committee of “Regione Calabria Comitato Etico Sezione Area Centro”, n.320, 21 December 2017. Written informed consent was obtained from the patients’ authorized representatives before study enrollment. 

### 2.2. Study Procedures

Each participant’s demographic information, comorbidities, prior medical history, and dietary condition, head and lumbar spine CT scan reports were recorded at the time of admission. Specifically, we included information about age, sex, etiology, time post-onset (days), presence of tracheal cannula, percutaneous endoscopic gastrostomy (PEG), nasogastric tube (NGT), previous decompressive craniectomy, level of consciousness as assessed by Coma Recovery Scale–Revised (CRS-R) and level of cognitive functioning (LCF); disability level through Disability Rating Scale (DRS) (Table 1). Next, a clinical follow-up at 2 months from admission during their IRU stay. The clinical examination was conducted again by a different clinician who was unaware of the study’s objectives.

#### Electrophysiological Screening

Severe DoC in patients prevents the assessment of voluntary muscle contraction, particularly foot dorsiflexion and eversion, even though it was possible to check and assess the anatomic localization of tibialis anterior muscle motor points and the course of the common peroneal nerve at the fibular head. Electrophysiological screening was performed by two skilled neurophysiopathologist technicians, blind to the study purpose, using a Cadwell EMG Sierra Summit (2023 Cadwell^®^ Industries Inc, (Kennewick, WA USA). In consideration of the purpose of the study, the recordings were conducted on the bilateral lower limbs [4,10,11,12,13]. The screening test included conduction studies of the two peroneal nerves. Compound muscle action potentials (CMAPs) were recorded from tibial anterior muscle. They were elicited by stimulating the peroneal nerve above and below the fibular head to diagnose demyelinating or compressive PN [4]. Sural nerve conduction studies were carried out but sensitive action potential (SAP) amplitude was not considered in the analysis of dates for their high range of variability due to technical factors (environmental electrical noise), pathophysiological conditions (tissue oedema) and frequent anatomic nerve course variants (inaccurate placement of recording electrodes). For peroneal motor nerve conduction, supramaximal stimulation was given to obtain the best CMAP amplitudes. Reduced CMAP amplitudes were defined when they were less than 2 mV (under the lower limit of the normal range for our laboratory). To reduce the complexity of the tests in IRU, we slightly modified the technique of direct muscle stimulation (DMS) [14,15,16,17]: a monopolar needle stimulation electrode was placed in the distal third of the tibialis anterior muscle with an anode positioned 3 cm distally. A monopolar needle recording electrode was placed 5 cm proximally with the reference positioned medially at the level of the tibial crest. The ground electrode was placed between the stimulating and recording monopolar needles (Figure 1). 

The tibialis anterior muscle was subjected to needle electromyography; however, the results were not considered during the data analysis phase of the study due to the challenge of assessing and measuring the morphology, parameters, and presence or absence of spontaneous activity in the motor unit due to the lack of voluntary recruitment in unconscious patients.

The dmCMAP was evoked first by direct muscle stimulation; the peroneal nerve was stimulated below the head of the fibula to obtain the neCMAP. The intensity of the direct muscle stimulation was gradually increased until supramaximal stimulation was achieved (intensity from 10 to 100 mA, duration from 0.1 to 1 ms). The bandwidth of the amplifier was between 2 Hz and 5 kHz (other test settings were: sweep: 2–5 ms/division; amplification: between 100 V and 10 mV/division). The screening tests were performed 3 to 5 times for each subject and the average values of the ratio between the peaks of the amplitudes obtained from nerve and muscle stimulation, respectively, were calculated. All recordings were reviewed for quality control and interpreted by two neurophysiologists with expert knowledge in electromyography. The investigators were blinded to the clinical and neurophysiological assessments at the time of their review and interpretation.

Direct muscle stimulation (DMS) in conjunction with NCS standard testing was the screening method used to distinguish CIP from CIM in patients with ABI. In our study, we used peroneal motor NCS alone without sural sensory NCS. If the ratio between neCMAP and mCMAP was closer to 1 (if both responses are absent, it is equal to 1) patients were classified as CIM (Figure 1), whereas if the ratio is <0.5, patients were classified as CIP (Figure 2).

### 2.3. Statistical Analysis

Univariate analyses were performed using SPSS (Statistical Package for Social Sciences, version 12.0, http://www.spss.it/; accessed on 1 January 2024). Assumptions for normality were tested for all continuous variables by using the Kolmogorov–Smirnov test. Firstly, the demographical and clinical data of the two groups were compared at admission to exclude significant differences before follow-up assessment. Considering the small sample size, non-parametric statistics (Mann–Whitney U-tests, Wilcoxon signed-rank test and χ^2^ test) were applied to analyze the effects of group and time. The Wilcoxon signed-rank test was used to evaluate differences in clinical data between time T0 and time T1 (2-months), to assess the presence of clinical worsening in the two groups. Clinical changes as measured by CRS-R, LCF and DRS were calculated as differential T1−T0 (delta) scores. All analyses had two-tailed alpha levels of 0.05 for defining significance. 

## 3. Results

During the study period, 28 patients transferred from the ICUs, or the neurosurgery units were evaluated. Twenty fulfilled the inclusion criteria and were included in the analysis (11 female; mean age: 55.1 ± 16.2; mean time-to-onset: 40.7 ± 20,34; mean CRS-R: 11.9 ± 6.32; mean LCF: 3.3 ± 1.30; mean DRS: 21.45 ± 3.33). Using electrophysiological screening, the diagnosis of unilateral PN was made in only 4 patients, CIP and/or CIM were present in 10 patients; only 6 patients resulted within the normal screening limits. No significant differences were detected at admission for any demographic or clinical variables between the CPINM group and other ABI patients (Table 1). Within the CIPNM group, electrophysiological screening enabled the separation of the two distinct patient categories with CIM and CIP (6 CIM and 4 CIP), but the limited sample size precluded separate data analysis (see Table 1). 

At follow-up, CIPNM patients tended to have a poor outcome. No CIPNM patients achieved cannula weaning with respect to admission, compared to 90% of other ABI patients (χ^2^ = 9.9; *p*-level = 0.001). According to CRS-R (CIPNM Δ: 3 [0–10]; NO CIPNM Δ: 6 [0–10]) and DRS scores (−2 [−1/−7]; −3.5 [−1/−7]) patients with CIPNM also had a lesser (nearly significant; *p*-level= 0.07 and 0.06, respectively) clinical recovery than patients without CIPNM (Table 1; Figure 3). In the CIPNM group, the CIM patients had the lowest clinical recovery rate (see Appendix A).

Due to the subacute rehabilitation setting of our study, it was not possible to evaluate the motor results of recovery of the standing position, functional walking and balance, impaired by the presence of unilateral PN.

## 4. Discussion

The diagnosis of mononeuropathy or polyneuropathy conventionally requires the execution of a formal electrodiagnostic study of the upper and lower limbs [5,6,7,8]. In addition, it is also necessary to perform repetitive nerve stimulation and needle electromyography to exclude neuromuscular junction disorders and myopathy [7,8]. However, the unconsciousness and lack of cooperation of patients represent serious obstacles to the completion of these tests. Therefore, in our study, we used a simplified electrophysiological screening to investigate weakness and atrophy of the foot dorsiflexor muscles (FDS) in patients with severe ABI; as a secondary objective, we aimed to investigate whether the resulting diagnosis (unilateral PN or CPNM) had a significant association with the rehabilitative outcomes. When compared to traditional diagnostic testing, peroneal NCS alone has already been shown to be the neurophysiological method with the best sensitivity and diagnostic accuracy [7,8]. In order to differentiate CIP from CIM, bilateral peroneal NCS was linked to bilateral DMS of the tibialis anterior muscle in our investigation. In 20% of cases, unilateral PN was diagnosed by including the electrophysiological screening test previously documented in the literature [7,8], but CIP and/or CIM were present in 50% of patients. The results of our investigation support previous findings in the literature [9,10,11,13], which differentiate between the CIP and CIM roles in FDS in patients with acquired brain injury [14,15,16,17]. Even if our study is restricted to a subacute rehabilitation context, only the early detection of patients with CIPNM is a critical goal since it has been linked to poor patient outcomes [5,6,7,8]. In fact, we showed that after two months of hospitalization, patients with CIPNM had greater difficulty (0%, out of 8 patients with tracheostomy at admission) weaning themselves from the endotracheal tube compared to ABI patients without CIPNM (90%, out of 10 patients with tracheostomy at admission). This finding agrees with previous studies suggesting that CIPNM is linked to a temporal delay in cannula weaning [5,6].

Similarly, we also detected a reduced recovery in the level of consciousness (CRS-rR) as well as disability level (DRS) in CIPNM, compared to ABI patients without CIPNM. Despite this, the finding did not reach a significant level (*p*-level= 0.07 and *p*-level = 0.06, respectively); however, some previous authors highlighted this evidence. Despite most of the literature focusing on the neurophysiological diagnosis, risk factors, prevention, and therapy related to CIPNM, only a small number of papers evaluated the potential impact of this comorbidity on the outcome of ABI patients. According to Hakiki et al. [5,6], ABI patients with CIPNM during their IRU stay required more time for decannulation and had a poorer chance of regaining oral nutrition and functional autonomy upon discharge. Intiso et al. [18] demonstrated that ABI patients with CIPNM are characterized by higher mortality and length of stay in IRU as well as by poor outcome. 

Our protocol was not able to assess the functional results pertaining to the recovery of standing, walking, and balance because of the subacute rehabilitation setting of our study, which may have been compromised by the occurrence of unilateral PN. Some other limitations related to our study need to be highlighted. First, we were unable to confirm the coexistence of neuromuscular junction diseases due to the non-performance of recurrent nerve stimulation [12]. Second, even though the CIPNM prevalence data are like those reported in the literature, ranging from 32.4% to 83.2%, for the various diagnostic criteria and mixed cases, we were unable to examine the sensitivity and diagnostic accuracy of simplified electrophysiological screening in comparison to complete electrophysiological tests. [5,6,7,8]. Third, because the study was conducted at a single hospital’s IRU, the population sample that was included was small and selective. For this reason, our evidence should be considered in the context of an exploratory study. 

## 5. Conclusions

Concomitant mononeuropathy, polyneuropathy or myopathy in patients with ABI are relevant comorbidities; nevertheless, often in rehabilitation settings, weakness, and atrophy of the dorsiflexion muscles of the feet are often undiagnosed, underestimating its potential to interfere with functional outcome [1,2,3]. The current study used a simplified electrophysiological screening technique to investigate the prevalence of PN or CIPNM in patients with ABI. We found that after extensive standard inpatient rehabilitation treatment for two months, ABI patients with CIPNM tended to have poor clinical outcomes. Patients with CIPNM should receive more intensive rehabilitation, with an emphasis on improving not only their level of consciousness and motor impairment but also their capacity to be weaned off of their endotracheal tube and resume oral nourishment. [7,15]. Future multicentric and multi-setting (ICU and IRU) research should validate the practical use and relevance of our electrophysiological assessment in the rehabilitative management of ABI patients.

## Figures and Tables

**Figure 1 biomedicines-12-00878-f001:**
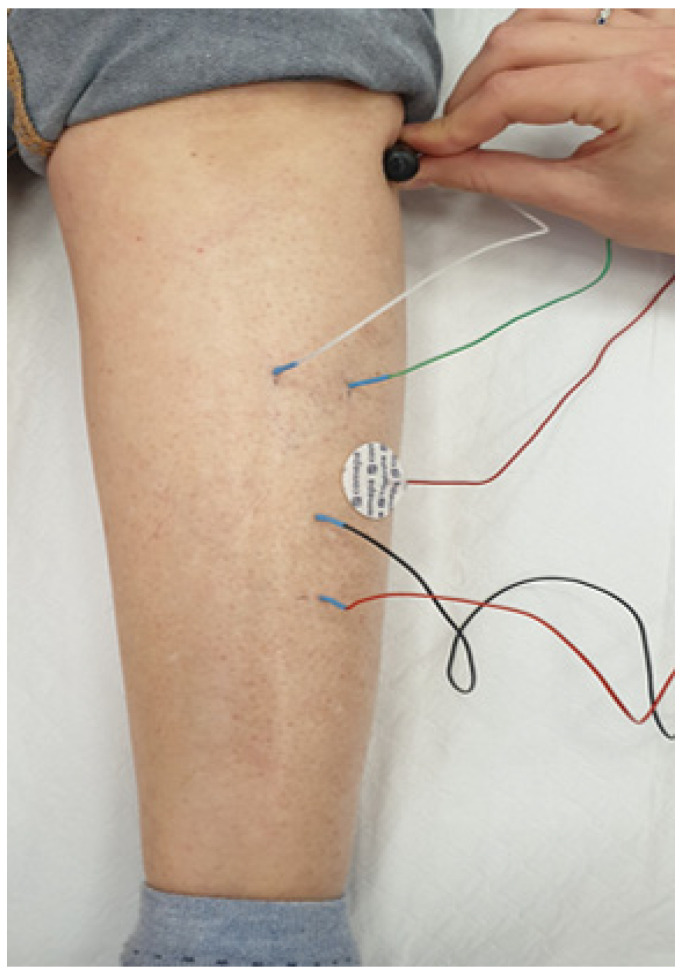
Modified technique of direct muscle stimulation (DMS) associated with motor conduction peroneal nerve across the fibular head (NCS).

**Figure 2 biomedicines-12-00878-f002:**
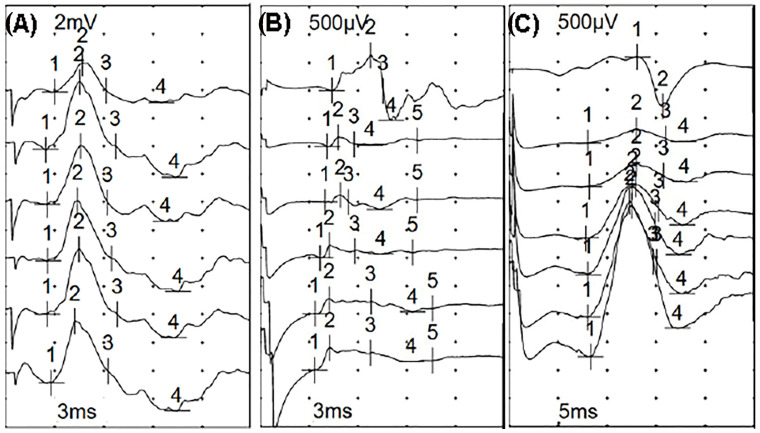
Electrophysiological amplitude response to DMS and NCS. (**A**) Normal amplitude responses to both nerve stimulation (1° trace) and direct muscle stimulation (2–6° traces). (**B**) CIM low amplitude responses to both nerve stimulation (1° trace) and direct muscle stimulation (2–6° traces). (**C**) CIP low amplitude response to nerve stimulation (1° trace) with respect to increasing amplitude responses to direct muscle stimulation (2–6° traces).

**Figure 3 biomedicines-12-00878-f003:**
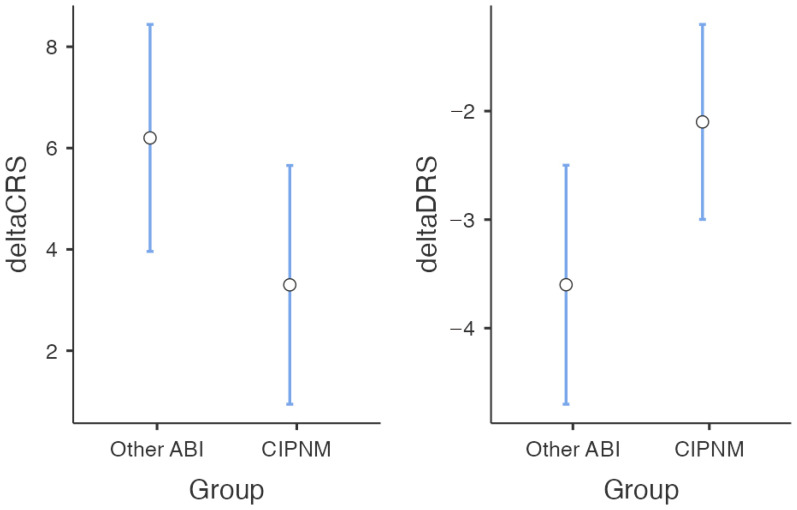
Worsening of clinical recovery in CIPNM group with respect to ABI patients without CIPNM at 2-month follow-up evaluation. Statistical analysis was performed on Δ scores between admission and follow-up assessment. Median and range values have been plotted.

**Table 1 biomedicines-12-00878-t001:** Demographical and clinical characteristics at admission in IRU and at follow-up.

	CIPNM (n° 10)	No CIPNM (n° 10)	*p*-Level
Age (years)	63 [41–75]	59.5 [18–74]	U = 43.5; *p*-level = 0.64
Sex (F/M)	6/4	5/5	χ^2^ = 0.2; *p*-level = 0.66
Traumatic etiology	2	2	n.s
Vascular etiology	8	8	n.s
Time post-onset (days)	48 [19–76]	29 [19–58]	U = 30.5; *p*-level = 0.15
** *Feeding* **			
PEG (% yes) at admission	3	3	n.s.
PEG (% yes) at 2 months	2	2	n.s.
NGT (% yes) at admission	4	7	χ^2^= 1.8; *p*-level = 0.15
NGT (% yes) at 2 months	2	2	n.s.
** *Tracheostomy* **			
% Yes at admission	8	10	χ^2^= 0.5; *p*-level = 0.48
% Yes at 2 months	8	1	** χ^2^ = 9.9; *p*-level = 0.001 **
** *Clinical scores* **			
CRS-R at admission	15 [3–23]	12 [4–23]	U = 43; *p*-level = 0.62
CRS-R at 2 months	18 [4–23]	18 [10–23]	U = 47; *p*-level = 0.84
Δ CRS-R at two months	3 [0–10]	6 [0–10]	** U = 26; *p*-level = 0.07 **
LCF at admission	4 [2–5]	3 [2–6]	U = 47.5; *p*-level = 0.87
LCF at 2 months	5 [3–7]	5 [3–7]	U = 41; *p*-level = 0.5
Δ LCF at two months	1 [0–3]	2 [0–3]	U 33.5; *p*-level = 0.21
DRS at admission	19 [17–26]	21.5 [17–26]	U = 45; *p*-level = 0.73
DRS at 2 months	18 [15–24]	18.5 [15–21]	U = 45; *p*-level = 0.71
Δ DRS at two months	−2 [−1/−7]	−3.5 [−1/−7]	** U 25.5; *p*-level= 0.06 **

Abbreviations: CIPNM: critical illness polyneuromyopathy; NGT: nasogastric tube; CRS-R, Coma Recovery Scale-Revised; LCF, level of cognitive functioning; DRS, disability rating scale; PEG: percutaneous endoscopic gastrostomy. Data are shown as median and range values. N.S.: not significant.

## Data Availability

The datasets associated with the present study are available upon reasonable request by interested researchers.

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
