# Peer review of "Electrophysiological Screening to Assess Foot Drop Syndrome in Severe Acquired Brain Injury in Rehabilitative Settings"

_biomedicines, 2024, doi:10.3390/biomedicines12040878_

Round 1

Reviewer 1 Report

Comments and Suggestions for Authors

I appreciate the authors for presenting this clinically useful research article. Although the sample size is small (20 patients), the results show positive findings that a simplified electrophysiological screening may allow for early identification of unilateral PN or CIPNM in severe ABI patients, thereby contributing to better functional prognosis in rehabilitative settings. My comments are as follows:

 1. In lines 200-202, the authors state, "The electrophysiological screening also allowed us to distinguish two specific categories of patients with CIM (n°6) and CIP (n°4) into the CIPNM group (see Table S1)." I am unsure if it is reasonable to combine these two groups (CIM and CIP) of patients into the CIPNM group. Please clarify this decision.

2. In lines 274-277, the authors concluded, "We found that after extensive standard inpatient rehabilitation treatment for two months, ABI patients with CIPNM tended to have poor clinical outcomes. This finding indicates that patients suffering from CIPNM could benefit from early and more intensive rehabilitation programs to obtain better clinical outcomes." I do not agree with the statement that patients suffering from CIPNM could benefit from early and more intensive rehabilitation programs to obtain better clinical outcomes. Please provide further clarification.

3. Diagnostic accuracy is a key factor in this study. Although the authors found that the simplified electrophysiological screening method shows promise, they should discuss its diagnostic accuracy compared to conventional diagnostic tests.

Author Response

Response to reviewer comments:

  1. In lines 200-202, the authors state, "The electrophysiological screening also allowed us to distinguish two specific categories of patients with CIM (n°6) and CIP (n°4) into the CIPNM group (see Table S1)." I am unsure if it is reasonable to combine these two groups (CIM and CIP) of patients into the CIPNM group. Please clarify this decision.

REPLY: Critical polyneuropathy (CIP) and myopathy (CIM) often coexist and are considered and studied as a combined clinical condition (CIPNM) by some authors (Piva et al. F1000Research. 2019; Hakiki et al. Acta Neurol Scand. 2020). Although electrophysiological screening allowed us to distinguish the two specific categories of patients, the small number of cases of CIM and CIP included in the study, did not allow us to analyze the data separately.

In the revised version we clarified the reason for the grouping of the two clinical conditions in lines 66-69 and in lines 201-204

  1. In lines 274-277, the authors concluded, "We found that after extensive standard inpatient rehabilitation treatment for two months, ABI patients with CIPNM tended to have poor clinical outcomes. This finding indicates that patients suffering from CIPNM could benefit from early and more intensive rehabilitation programs to obtain better clinical outcomes." I do not agree with the statement that patients suffering from CIPNM could benefit from early and more intensive rehabilitation programs to obtain better clinical outcomes. Please provide further clarification.

REPLY: According to the previous literature (Hakiki et al. Acta Neurol Scand. 2020; Hakiki et al., Diagnostics 2022), patients with CIPNM require more time for weaning from ventilatory support in the ICU and for decannulation and acquisition of oral nutrition in the IRU. Our study, conducted in a subacute rehabilitation setting, suggests that rehabilitative treatments in patients suffering from CIPNM should be intensified and focused not only on the improvement of the level of consciousness and motor disability but also on advancing weaning from the endotracheal tube and recovery of oral nutrition.

In the revised version we inserted this argumentation in lines 282-284

  1. Diagnostic accuracy is a key factor in this study. Although the authors found that the simplified electrophysiological screening method shows promise, they should discuss its diagnostic accuracy compared to conventional diagnostic tests.

REPLY: Peroneal NCS alone has already been identified as the neurophysiological technique with the best sensitivity and diagnostic accuracy compared to conventional diagnostic tests (Latronico et al., Lancet Neurol. 2011; Zipko et al., J Neurol. Sci.1998 ). In our study, bilateral peroneal NCS was associated with bilateral DMS of the tibialis anterior muscle, to distinguish CIP from CIM. By integrating the electrophysiological screening test, already reported in the literature, the diagnosis of unilateral PN was made in 20% of cases, while CIP and/or CIM were present in 50% of patients. The data from our study confirm what has been reported in the literature, distinguishing the contribution of CIP or CIM to FDS in ABI patients.

 In the revised version we inserted this argumentation in lines 235-243 of the Discussion Section

Reviewer 2 Report

Comments and Suggestions for Authors

This study investigates the application of electrophysiological screening in ABI patients for detection of critical polyneuropathy or critical mypathy and peroneal neuropathy.

The Introduction is very clearly written and provides a succinct and informative background to PN and CIP.

The Methods section is also very clearly written. 

The Results are clear with good use of tables and figures. 

Line 208: Change "gastrostostomy" to "gastrostomy."

Figure 3: What do the squares indicate?

Line 196: This sentence is unclear. was it ten or six patients within the normal limits?

Line 230: Are you looking at hypertrophy or atrophy or either?

Line 233: Change "compared to" to "compared with."

Line 256: Change the start of the sentence. "It was not able..." What was not able? "Our protocol..." or "Our test procedure..." or "This method..." for example. 

Line 270: Again you mention "weakness and hypertrophy..." I think you mean "weakness and atrophy."

Author Response

Reviewer n°2

  • The Introduction is very clearly written and provides a succinct and informative background to PN and CIP. The Methods section is also very clearly written. The Results are clear with good use of tables and figures. 

REPLY: We express our gratitude to the reviewer for their thoughtful and insightful feedback.

  • Line 208: Change "gastrostostomy" to "gastrostomy."

REPLY: Done

  • Figure 3: What do the squares indicate?

REPLY: Median and range values have been plotted.

  • Line 196: This sentence is unclear. was it ten or six patients within the normal limits?

REPLY: Done

  • Line 230: Are you looking at hypertrophy or atrophy or either?

REPLY: Atrophy

  • Line 233: Change "compared to" to "compared with."

REPLY: Done

  • Line 256: Change the start of the sentence. "It was not able..." What was not able? "Our protocol..." or "Our test procedure..." or "This method..." for example. 

REPLY: Done

  • Line 270: Again you mention "weakness and hypertrophy..." I think you mean "weakness and atrophy."

REPLY: Done

Round 2

Reviewer 1 Report

Comments and Suggestions for Authors

The authors have replied my comments item-by-item. I have no more comments. Accept is my final decision.